# Dysregulated Dscam levels act through Abelson tyrosine kinase to enlarge presynaptic arbors

Gabriella R Sterne[1,2†], Jung Hwan Kim[1,2†], Bing Ye[1,2*]

[1]Life Sciences Institute, University of Michigan, Ann Arbor, United States;
[2]Department of Cell and Developmental Biology, University of Michigan, Ann Arbor, United States

**Abstract** Increased expression of Down Syndrome Cell Adhesion Molecule (Dscam) is implicated in the pathogenesis of brain disorders such as Down syndrome (DS) and fragile X syndrome (FXS). Here, we show that the cellular defects caused by dysregulated Dscam levels can be ameliorated by genetic and pharmacological inhibition of Abelson kinase (Abl) both in Dscam-overexpressing neurons and in a *Drosophila* model of fragile X syndrome. This study offers Abl as a potential therapeutic target for treating brain disorders associated with dysregulated Dscam expression.

*For correspondence: bingye@umich.edu

†These authors contributed equally to this work

## Introduction

Dscam levels are increased in the brains of human patients with DS and in mouse models of DS (*Saito et al., 2000*; *Alves-Sampaio et al., 2010*). Recent research also suggests that fragile X mental retardation protein (FMRP) binds directly to the mRNAs of Dscam from mouse brain (*Brown et al., 2001*; *Darnell et al., 2011*), and studies in *Drosophila* neurons further confirmed that FMRP suppresses Dscam translation (*Cvetkovska et al., 2013*; *Kim et al., 2013*). In the dendritic arborization (da) neurons in *Drosophila* larva, Dscam expression level is instructive for presynaptic terminal growth (*Kim et al., 2013*). Consistent with this, increased Dscam in *Drosophila* FXS models results in enlarged presynaptic arbors (*Kim et al., 2013*). These findings indicate the importance of proper Dscam levels in normal development and in the pathogenesis of brain disorders.

Because of the link between increased Dscam expression and neuronal defects in DS and FXS models, targeting Dscam or its signaling mechanism might prove therapeutic for these disorders. Currently, neither methods for targeting Dscam proteins nor those for targeting the signaling pathway activated by dysregulated Dscam are available, impeding the development of such therapies. In fact, very little is known about how Dscam signaling is transduced in vivo. In *Drosophila*, Dscam has previously been shown to bind to Dock (*Schmucker et al., 2000*), while in mammals it has been shown to associate with Uncoordinated-5C, Focal adhesion kinase (FAK), Fyn kinase, and PAK1 (*Li and Guan, 2004*; *Purohit et al., 2012*). In addition, studies suggest possible genetic interactions between Dscam and the Abelson tyrosine kinase (Abl) in neurite development in the central nervous system (CNS) of *Drosophila* embryos (*Andrews et al., 2008*; *Yu et al., 2009*). However, evidence demonstrating the requirement of these potential interactors for the defects that arise from increased Dscam expression is lacking. Moreover, whether pharmacologically targeting these molecules in vivo might alleviate the effects of increased Dscam expression is unknown.

The evolutionarily conserved Abl kinase transduces extracellular cues into cytoskeletal rearrangements that affect cell motility and shape (*Bradley and Koleske, 2009*) and is implicated in axonal development, including axon guidance and extension (*Wills et al., 1999a*; *Wills et al., 1999b*; *Wills et al., 2002*; *Hsouna et al., 2003*; *Lee et al., 2004*; *Forsthoefel et al., 2005*). Overexpression of Abl

**eLife digest** Information is transmitted through the brain by cells called neurons, which are connected into specific circuits and networks. As the brain develops, several different signaling molecules control how the connections between neurons develop. If these signals occur at the wrong time or wrong place, or in the wrong amount, the neurons may not connect in the right way; this is the cause of several brain disorders.

One of the signaling molecules that helps neural circuits to form in the developing brain is the Dscam protein. Having too much Dscam has been linked to neurons with enlarged presynaptic terminals. Presynaptic terminals are the parts of each neuron that send information on to the next cell, and when they are enlarged it results in the neuron not being able to communicate with other neurons in a targeted way. People with brain disorders including Down syndrome, epilepsy and possibly fragile X syndrome often have excessive amounts of Dscam.

It was not known precisely how Dscam signals within neurons. Sterne, Kim and Ye have now investigated this by exploring the effects of Dscam on a group of well-known neurons in the larvae of the fruit fly species *Drosophila*. The presynaptic terminals of single neurons in this group were labeled in the larvae using a genetic marker. This revealed that the neurons of larvae that had been engineered to produce too much Dscam had larger presynaptic terminals than normal.

Further investigation showed that for Dscam to influence how a presynaptic terminal grows, it must interact with another signaling protein called Abelson tyrosine kinase (or Abl for short). Therefore, the larger presynaptic terminals seen in larvae that produce too much Dscam are a result of the Dscam protein activating too much Abl.

There are several drugs that are approved for use in humans that suppress the activity of Abl. Sterne, Kim and Ye used two of these to treat fruit fly larvae, and found that they reversed the detrimental effects of extra Dscam on the larvae's neural circuit. Furthermore, the drugs fixed neural defects in a fruit fly model designed to reproduce the symptoms of fragile X syndrome.

Overall, the results presented by Sterne, Kim and Ye suggest that suppressing the abnormally high activity of the Abl protein could be a way of treating the brain disorders caused by having excessive amounts of the Dscam protein. The next step is to test whether Dscam and Abl interact in the same way in mammals and whether the proposed treatment is effective in treating mammalian models of disorders that involve dysregulated Dscam signaling.

causes increased axon growth in the *Drosophila* CNS (*Leyssen et al., 2005*), which is reminiscent of the effect caused by Dscam overexpression in C4da neurons (*Kim et al., 2013*). In addition, previous studies in *Drosophila* have indicated that *abl* mutations have an additive effect with *Dscam* mutations, such that *abl/Dscam* double mutant embryos have more severe axon midline crossing defects than either *abl* or *Dscam* mutants alone (*Andrews et al., 2008*; *Yu et al., 2009*). However, the molecular nature of this interaction, that is, whether or not Dscam acts through Abl, and particularly whether inhibition of Abl mitigates neuronal defects caused by dysregulated Dscam, is unknown.

Here we show that Dscam activates Abl through its cytoplasmic domain, which is required for the presynaptic arbor enlargement caused by dysregulated Dscam expression in vivo. Importantly, we demonstrate that the pharmacological inhibition of Abl ameliorates exuberant presynaptic arbor growth both in flies overexpressing Dscam and in a fly model of FXS.

## Results and discussion

We took advantage of the *Drosophila* larval class IV dendritic arborization (C4da) neurons to delineate the molecular mechanism of Dscam signaling in presynaptic arbor development, because the presynaptic terminal growth of these neurons is highly sensitive to Dscam levels in a linear fashion (*Kim et al., 2013*). For example, loss of *Dscam* causes C4da presynaptic terminals to fail to grow while increased Dscam levels lead to increased presynaptic terminal growth (*Kim et al., 2013*). From tests of candidate genes that potentially mediate Dscam function, including FAK, Fyn, PAK, RhoA, and Abl, we identified Abl as a key molecule mediating Dscam's functions in presynaptic terminal growth. We first asked whether Abl is sufficient to promote presynaptic terminal growth in C4da neurons. Consistent with a previous study performed in *Drosophila* adult CNS neurons (*Leyssen et al., 2005*),

overexpression of Abl in C4da neurons caused significant overgrowth of the presynaptic terminals (*Figure 1A,B,E*). Since Abl is known to have both kinase-dependent and kinase-independent functions (*Henkemeyer et al., 1990*; *Schwartzberg et al., 1991*; *Tybulewicz et al., 1991*), we tested whether expression of a kinase-dead form of Abl, Abl-K417N (*Henkemeyer et al., 1990*; *Wills et al., 1999b*), could promote presynaptic terminal growth. We found that C4da presynaptic terminals over-expressing Abl-K417N were indistinguishable from wild-type (*Figure 1D,E*), indicating that Abl kinase activity is required. Consistent with the idea that Abl kinase activation is important, expression of a constitutively active form of Abl, BCR-Abl, led to extremely exuberant overgrowth (*Figure 1C,E*). Taken together, these results suggest that Abl is sufficient to promote presynaptic terminal growth and that the extent to which Abl instructs presynaptic terminal growth is related to Abl kinase activation.

Since overexpression of Abl increases presynaptic terminal growth, similar to Dscam, we next tested whether Dscam requires Abl to instruct presynaptic terminal growth. For this, we used the mosaic analysis with a repressible cell marker (MARCM) technique to overexpress Dscam in *abl¹* mutant C4da neurons (*Lee and Luo, 2001*) and assessed presynaptic terminal length. We found that although Dscam overexpression led to significantly (150%) longer presynaptic terminals than wild-type clones (*Figure 1F,G,O*), *abl¹* mutant clones that overexpressed Dscam did not differ in length from *abl¹* mutant clones (*Figure 1H,I,O*). Presynaptic terminal length was also subtly but significantly shorter in *abl¹* mutant clones compared to wild-type controls (*Figure 1I,O*). A different loss-of-function allele of *abl*, *abl⁴*, exhibited similar effects on the presynaptic overgrowth caused by Dscam overexpression (*Figure 1J,K,O*), confirming that loss of *abl* function is responsible for blocking the presynaptic phenotypes caused by increased Dscam levels. As a control, *abl* loss-of-function mutations did not affect the expression of the Dscam transgenes in the C4da cell body or presynaptic terminals (*Figure 1—figure supplement 1*).

As a further proof-of-concept, we asked whether loss of *abl* could mitigate the effects of dysregulated Dscam levels without utilizing Dscam transgenes. FXS is caused by an absence of FMRP (*Kremer et al., 1991*), and is modeled in *Drosophila* using loss-of-function mutants for the *Drosophila* homolog of *FMR1*, *dFMRP* (*Zhang et al., 2001*; *Dockendorff et al., 2002*). It has previously been shown that FMRP binds to Dscam mRNA in both mammals and *Drosophila* (*Darnell et al., 2011*; *Cvetkovska et al., 2013*; *Kim et al., 2013*) and that dFMRP represses Dscam expression to control presynaptic terminal growth, so that *dFMRP* mutants exhibit increased presynaptic terminal length in C4da neurons (*Kim et al., 2013*). Strikingly, loss of only a single copy of *abl* significantly rescued presynaptic terminal length to wild-type levels (*Figure 1L–N,P*). These results suggest that Abl is required for Dscam to instruct presynaptic terminal growth.

An important function of Dscam in neuronal development is to mediate self-avoidance between neurites of the same neuron (*Zipursky and Grueber, 2013*). Abl does not seem to be required by Dscam for either dendrite growth (*Figure 1—figure supplement 2*) or for dendritic self-avoidance in C4da neurons. Loss of *abl* did not compromise the ectopic avoidance caused by overexpressing Dscam in distinct types of da neurons (*Hattori et al., 2007*; *Hughes et al., 2007*; *Matthews et al., 2007*) (*Figure 1—figure supplement 3*). This suggests a divergence in Dscam signaling for the development of presynaptic terminals and dendritic branches. Taken together, these results indicate that Abl is specifically required for Dscam-mediated presynaptic terminal growth.

Next, we asked how Abl might mediate Dscam signaling. Abl can be activated by binding to specific proteins, such as the cytoplasmic domains of membrane receptors (*Bradley and Koleske, 2009*). In contrast to the exuberant presynaptic terminal overgrowth caused by Dscam overexpression in C4da neurons (*Figure 2A*, middle), overexpressing a mutant form of Dscam that lacked most of the cytoplasmic domain (DscamΔCyto) did not cause presynaptic terminal overgrowth (*Figure 2A*, bottom). DscamΔCyto was trafficked to the axon terminals and expressed at a similar level to full-length Dscam (*Figure 2—figure supplement 1*). These results suggest that the cytoplasmic domain is required for Dscam to instruct presynaptic terminal growth.

We then asked whether Dscam and Abl physically interact through the Dscam cytoplasmic domain. We found that Dscam and Abl proteins co-immunoprecipitated from transfected *Drosophila* Schneider 2 (S2) cells expressing these two proteins (*Figure 2B*, second lane from right). In contrast, DscamΔCyto did not co-immunoprecipitate Abl (*Figure 2B*, furthest right lane). These results suggest that Dscam and Abl proteins form a complex through Dscam's cytoplasmic domain. Next, to test the in vivo interaction of Dscam and Abl in presynaptic terminals specifically, we determined whether Abl

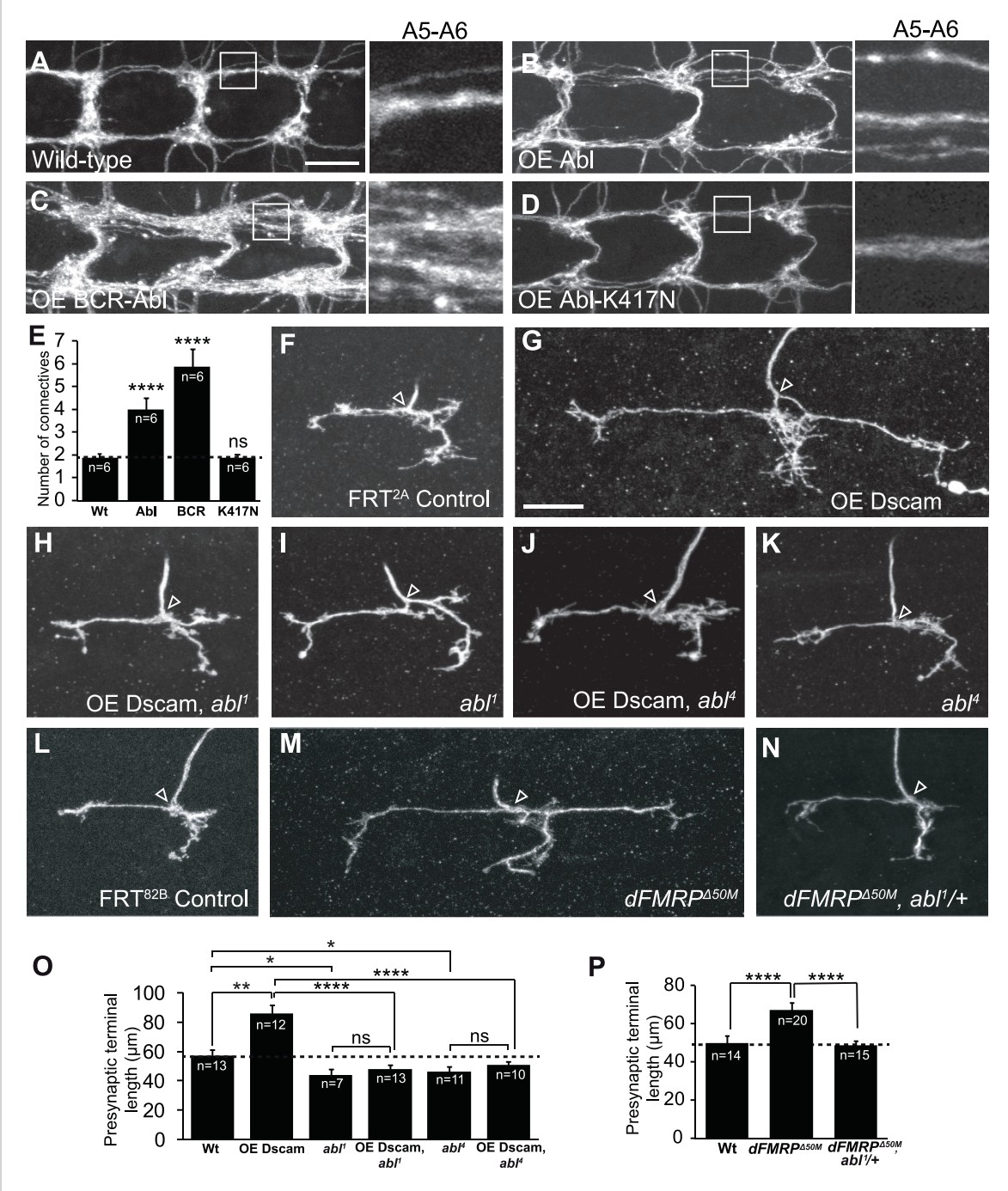

**Figure 1**. Dscam requires Abl to promote presynaptic terminal growth. (**A–E**) Abl is sufficient to cause presynaptic terminal overgrowth in C4da neurons. Transgenes were expressed with a C4da neuron-specific Gal4 driver, *ppk*-Gal4, and presynaptic terminals were visualized with a membrane monomeric RFP (mCD8-mRFP) transgene. Overexpression of Abl (**B**) leads to a modest increase in presynaptic terminal growth as compared to control (**A**). Overexpression of the constitutively active BCR-Abl (**C**) leads to robustly increased presynaptic terminal growth, while overexpression of kinase-dead Abl-K417N (**D**) is indistinguishable from control. Quantification of the number of axon connectives is shown in (**E**). Scale bar is 10 µm. (**F–K**) Abl is required in C4da neurons for Dscam to instruct presynaptic terminal growth. The arrowhead in each panel points to the location where an axon elaborates the presynaptic terminal arbor. The MARCM technique was used to generate and visualize single mutant C4da neurons. While overexpression of Dscam::GFP (**G**) in single C4da presynaptic terminals leads to increased length when compared to control (**F**), overexpression of Dscam in *abl¹* mutant neurons (**H**) leads to presynaptic terminal lengths that are indistinguishable from *abl¹* mutant neurons (**I**). Similarly, overexpression of Dscam in *abl⁴* mutant neurons (**J**) does not significantly change presynaptic terminal length when compared to *abl⁴* mutant neurons (**K**). (**L–N**) Abl is required to instruct presynaptic terminal growth in *dFMRP* mutants. (**M**) Loss of *dFMRP* leads to increased presynaptic terminal growth, which has previously been shown to require Dscam. Loss of

*Figure 1. continued on next page*

Figure 1. Continued

one copy of *abl* in *dFMRP^Δ50M* mutant neurons (**N**) leads to presynaptic terminal lengths that are indistinguishable from control (**L**). Scale bar is 10 μm. (**O** and **P**) Quantification of the presynaptic terminal length in C4da neurons of indicated genotypes. Sample number is shown in white within each bar.

The following figure supplements are available for figure 1:

**Figure supplement 1**. Loss of *abl* does not affect Dscam::GFP expression level.

**Figure supplement 2**. Loss of *abl* does not affect C4da dendritic length or morphology.

**Figure supplement 3**. Single Dscam isoform-induced ectopic repulsion between class I and class III dendrites does not require *abl*.

localization in presynaptic terminals was altered by the expression of Dscam or DscamΔCyto (*Figure 2C*). When expressed alone or with DscamΔCyto::GFP, Abl::Myc was diffusely distributed in the presynaptic terminals, with little colocalization with DscamΔCyto::GFP (*Figure 2C*, middle and bottom). However, when expressed with Dscam::GFP, Abl::Myc became more punctate and clearly colocalized with Dscam::GFP (*Figure 2C*, top). We used Manders' Correlation Coefficients to quantify the colocalization of Dscam::GFP and Abl::Myc. Colocalization analysis revealed a significant increase in both $M_1$ and $M_2$ (*Figure 2C*, bottom right) when Abl::Myc was coexpressed with Dscam::GFP as compared to when Abl::Myc was coexpressed with DscamΔCyto::GFP, where $M_1$ represents the fraction of Abl that overlaps with Dscam, and $M_2$ represents the fraction of Dscam that overlaps with Abl. These findings support the idea that Abl and Dscam interact in presynaptic terminals in vivo.

Do increased Dscam levels activate Abl kinase? In mammals, autophosphorylation of Abl at tyrosines 245 and 412 (Y245 and Y412) stabilizes the active conformation of the kinase (*Brasher and Van Etten, 2000*; *Tanis et al., 2003*). As a result, phospho-specific antibodies raised against Y412 have been employed to detect active Abl kinases (*Brasher and Van Etten, 2000*). This approach has been used successfully to recognize the phosphorylation of the corresponding tyrosines (Y539/522) in *Drosophila* as an assay for Abl kinase activation (*Stevens et al., 2008*). Since the ability of Abl to instruct presynaptic terminal growth relies on Abl kinase activity, we tested whether Dscam activates Abl using a phosho-Y412-Abl (p-Abl) antibody. We found that Abl kinase activation was significantly increased (2.6 fold) when Abl and Dscam were co-expressed in S2 cells (*Figure 3A*). Furthermore, unlike wild-type Dscam, DscamΔCyto did not increase Abl kinase activation. In fact, it appears to act as a dominant-negative, as Abl activity was significantly decreased from control (*Figure 3A*, right). As a negative control, no signal was detected when the kinase-dead Abl-K417N was blotted with p-Abl antibody in the same assay, suggesting that our assay specifically reported Abl activation (*Figure 3—figure supplement 1*). These results suggest that Dscam enhances Abl kinase activity. To investigate whether the same is true in presynaptic terminals in vivo, we devised a novel method of reporting Abl activation specifically in C4da presynaptic terminals. To achieve this, we used a previously reported probe that reports Abl activity, Pickles2.31 (*Mizutani et al., 2010*). Pickles2.31 is composed of a fragment of a characteristic Abl substrate, CrkL, sandwiched between the fluorescent proteins Venus and enhanced CFP (ECFP) (*Figure 3B*). It has previously been reported that activated Abl phosphorylates Pickles2.31 on the Y207 residue of the CrkL fragment, which can be detected with an antibody against CrkL-phospho-Y207 (p-CrkL) (*Mizutani et al., 2010*). After expressing Pickles2.31 specifically in C4da neurons with the *ppk*-Gal4 driver, we dissected the larval CNS and immunoprecipitated Pickles2.31 from the lysates. Since the cell bodies of C4da neurons reside in the body wall, using only the larval CNS allowed us to monitor Pickles2.31 phosphorylation only in the C4da neuron presynaptic terminals (*Figure 3C*). We found that overexpression of Dscam in C4da neurons led to an increase in Y207 phosphorylation of Pickles2.31 in the presynaptic terminals, while overexpression of DscamΔCyto was indistinguishable from control (mCD8-mRFP) (*Figure 3D*). Consistent with the notion that Pickles2.31 is an Abl activity indicator, overexpression of BCR-Abl led to a robust increase in phospho-Y207 levels as compared to the control. These results suggest that Dscam activates Abl both in culture and in C4da presynaptic terminals in vivo, and that this activation requires the cytoplasmic domain of Dscam.

These results raised the interesting possibility that targeting Abl might be a viable therapy for brain disorders caused by increased Dscam expression. Abl is a well-established target for treating chronic

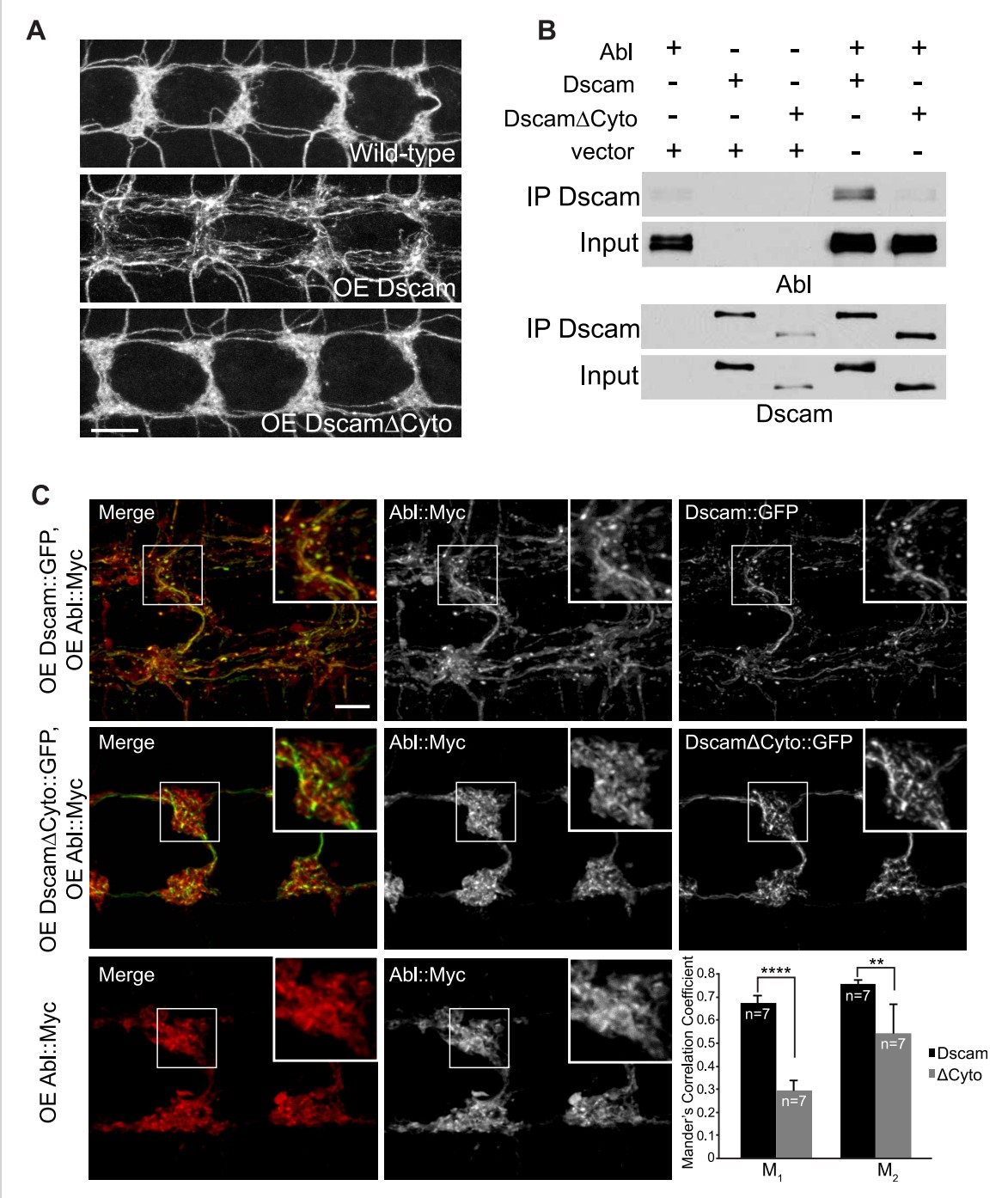

**Figure 2**. Dscam binds to Abl through its cytoplasmic domain. (**A**) The cytoplasmic domain of Dscam is required for instructing presynaptic terminal growth. Overexpression of full-length Dscam under the control of *ppk*-Gal4 (**A**, middle) leads to exuberant presynaptic terminal overgrowth when compared to control (**A**, top). However, overexpression of DscamΔCyto (**A**, bottom) fails to increase presynaptic terminal growth. Scale bar is 10 µm. (**B**) Dscam binds Abl via its cytoplasmic domain. S2 cells were co-transfected with Abl::Myc along with either Dscam::GFP, DscamΔCyto::GFP, or an empty vector. Dscam::GFP was immunoprecipitated with anti-GFP antibody and bound Abl::Myc was examined with anti-Myc antibody (top). Immunoprecipitated Dscam::GFP and input Dscam::GFP was examined with anti-GFP (bottom). (**C**) Abl colocalizes and redistributes with Dscam but not with DscamΔCyto in presynaptic terminals in vivo. When expressed alone, Abl::Myc shows a diffuse pattern (bottom). When expressed along with Dscam::GFP (top), Abl::Myc redistributes into punctate structures that colocalize with Dscam::GFP. When expressed along with DscamΔCyto::GFP (middle), Abl::Myc does not redistribute, displaying a similar pattern to when Abl::Myc is expressed alone (bottom). This is quantified using Manders' Correlation Coefficient. $M_1$ presents a measure of the fraction of Abl::Myc that overlaps with Dscam(ΔCyto)::GFP, while $M_2$ presents a measure of the fraction of

*Figure 2. continued on next page*

*Figure 2. Continued*

Dscam(ΔCyto)::GFP that overlaps with Abl::Myc. Both $M_1$ and $M_2$ are significantly increased in Abl-Dscam coexpression when compared to Abl-DscamΔCyto coexpression. Scale bar is 5 µm.
The following figure supplement is available for figure 2:

**Figure supplement 1**. DscamΔCyto::GFP is trafficked to presynaptic terminals at a similar level to Dscam::GFP.

myeloid leukemia, and there are multiple Abl inhibitors that are approved by the US Food and Drug Administration (FDA). As a proof-of-concept experiment, we attempted to rescue the developmental defects caused by Dscam overexpression using Abl inhibitors. We first tested nilotinib, which is a FDA-approved second-generation Abl kinase inhibitor that can cross the blood–brain barrier (*Weisberg et al., 2005*; *Hebron et al., 2013*). Using cultured S2 cells overexpressing Abl, we found that nilotinib robustly inhibited *Drosophila* Abl (*Figure 4A*). Based on these results, we tested whether administration of nilotinib to developing larvae could rescue the effects of increased Dscam expression in C4da presynaptic terminals in vivo. To do this, we performed MARCM to visualize single C4da neurons in animals fed nilotinib or vehicle and assessed presynaptic terminal length. While overexpression of Dscam caused increased (152%) presynaptic terminal length in animals fed vehicle (*Figure 4B–D*), the effect was significantly rescued (to 115% of control) by feeding the animals with nilotinib (*Figure 4B,E*). Consistent with the idea that these effects were due to inhibition of Abl activity rather than a reduction in Dscam expression, nilotinib did not change the expression of the Dscam transgene (*Figure 4—figure supplement 1A*).

Administration of nilotinib to developing larvae did not lead to adverse effects on overall development and neuronal growth. At the dose we used, nilotinib did not cause a change in presynaptic terminal growth (*Figure 4F*) or dendritic growth (*Figure 4—figure supplement 2A,B*) in wild-type larvae. Moreover, it did not impact the number of adults that eclosed or the dynamics of eclosion when compared to vehicle-fed flies (*Figure 4—figure supplement 2C,D*).

Although frequently used to inhibit pathological increases in Abl activity in patients, nilotinib is known to have several off-targets, including c-Kit, PDGFR, Arg, NQO2, and DDR1 (*Hantschel et al., 2008*). Consistent with the idea that nilotinib acts on Abl rather than on an off-target molecule to rescue presynaptic terminal growth, administering nilotinib to larvae overexpressing Dscam in $abl^1$ clones did not lead to a further decrease in presynaptic terminal length when compared to vehicle-fed control (*Figure 4—figure supplement 3C,D,G*). To further rule out the possibility that the observed rescue of presynaptic terminal length by nilotinib was the result of an off-target effect, we tested bafetinib, another Abl inhibitor with non-overlapping off-targets, Fyn and Lyn (*Kimura et al., 2005*). Bafetinib has also been shown to cross the blood brain barrier (*Santos et al., 2010*). Like nilotinib, administration of bafetinib to Dscam-overexpressing larvae led to a significant decrease in presynaptic terminal length (*Figure 4—figure supplement 3A,B,F,G*) without changing the expression of the Dscam transgene (*Figure 4—figure supplement 1B*). Bafetinib alone did not change presynaptic terminal length in wild-type larvae when compared to wild-type larvae fed vehicle (*Figure 4—figure supplement 3E,G*). Taken together, these results suggest that pharmacological inhibition of Abl mitigates the consequences of increased Dscam signaling in vivo.

We next sought to test the efficacy of nilotinib treatment in a model of a disease associated with dysregulated Dscam expression, FXS. Thus, we tested whether administration of nilotinib could rescue the presynaptic overgrowth caused by increased Dscam expression in *dFMRP* mutants. We found that, while *dFMRP* mutants fed vehicle showed a significant increase (130%) in presynaptic terminal length (*Figure 4B,G*), administration of nilotinib to *dFMRP* mutants almost completely rescued (to 103% of control) the exuberant presynaptic terminal growth to wild-type levels (*Figure 4B,H*). These results suggest that pharmacological inhibition of Abl kinase is effective for mitigating the effects of increased Dscam level in an in vivo model of FXS.

In this study, we show that Dscam requires Abl to promote presynaptic terminal growth in vivo and that the binding of Abl to the Dscam cytoplasmic domain leads to Abl kinase activation. Furthermore, we show that treating larvae with Abl inhibitors rescues the developmental defects caused by increased Dscam levels in vivo in both Dscam-overexpressing neurons and disease-relevant models. Taken together, these results suggest that Abl is a potential drug target for the treatment of brain disorders associated with dysregulated Dscam expression, including DS and FXS.

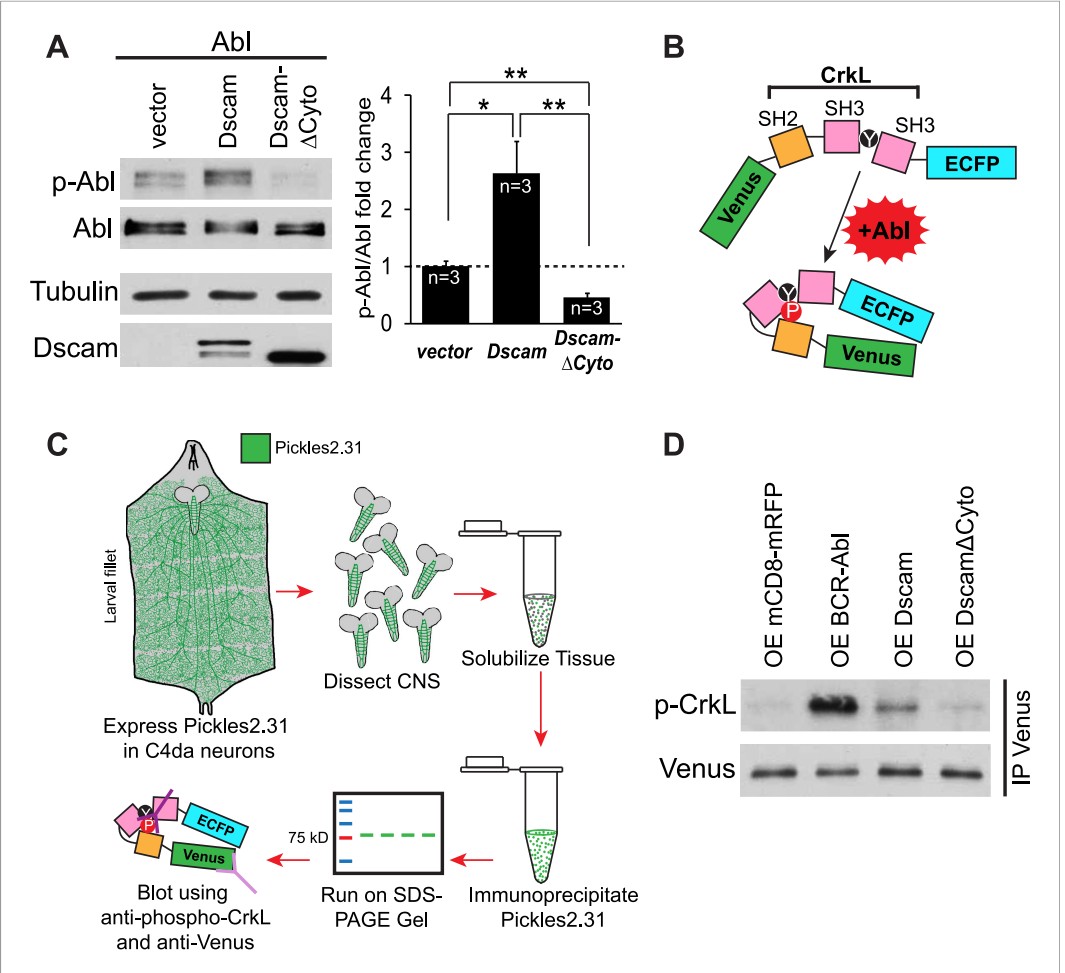

**Figure 3**. Dscam activates Abl kinase in culture and in vivo. (**A**) Dscam activates Abl in cultured S2 cells. Abl activation was examined in S2 cell lysates transfected with indicated constructs by using anti-phospho-Y412-Abl antibody. The intensity of phospho-Abl was quantified, normalized to total Abl::Myc, and presented as bar graph (n = 3) (**A**, right). (**B**) Schematic of Pickles2.31, an Abl activity reporter that uses phosphorylation of CrkL to report Abl kinase activity. Pickles2.31 is composed of a fragment of human CrkL that contains an Abl phosphorylation site, Y207, sandwiched between ECFP and Venus. Phosphorylation of Pickles2.31 by Abl can be detected with an anti-phospho-Y207-CrkL (p-CrkL) antibody. (**C**) Schematic of in vivo assay for detecting Abl activity in C4da presynaptic terminals. Pickles2.31 is specifically expressed in C4da neurons. As can be appreciated from the larval fillet diagram (left), the cell bodies and dendrites of C4da neurons reside in the larval body wall while their presynaptic terminals reside in the CNS. To assay Abl activity only in presynaptic terminals, larval CNS are dissected out and solubilized into lysates. Pickles2.31 in the presynaptic terminals is then immunoprecipitated with an anti-Venus antibody (left). After running on an SDS-PAGE gel, Pickles2.31 expression level can be assayed using an anti-Venus antibody, while the phosphorylation of Y207, a proxy for Abl activity level, can be ascertained by western blotting with a p-CrkL antibody. (**D**) Dscam activates Abl in presynaptic terminals in vivo. Overexpression of BCR-Abl leads to a robust increase in p-CrkL staining of Pickles2.31 when compared to the mCD8-mRFP control. Similarly, overexpression of Dscam leads to consistent, though less extreme, increase in p-CrkL when compared to control. In contrast, overexpression of DscamΔCyto is indistinguishable from the mCD8-mRFP control. This is a representative blot of three experimental repeats.

The following figure supplement is available for figure 3:

**Figure supplement 1**. Phospho-Y412-Abl antibody specifically reports Abl activation.

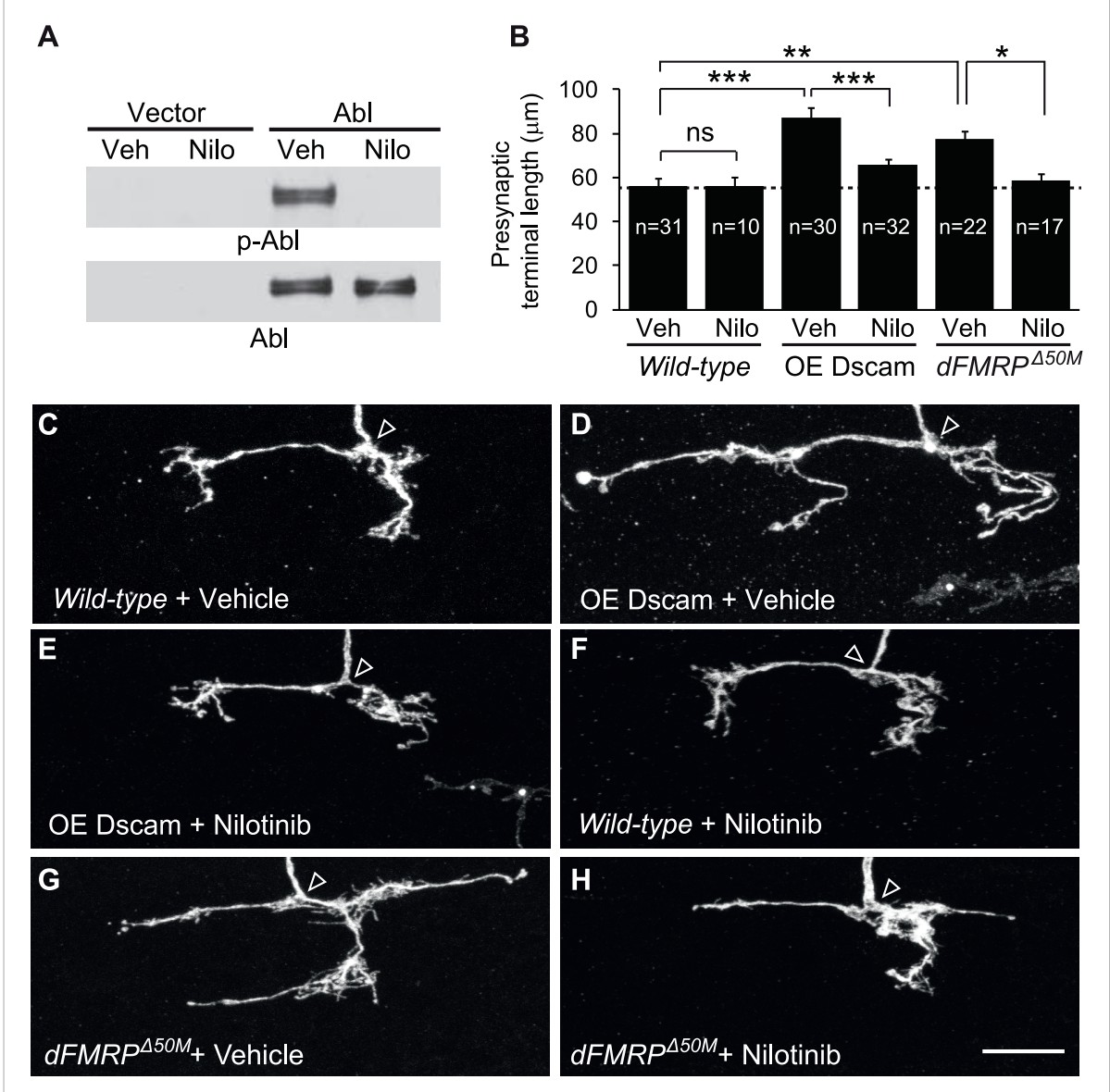

Figure 4. Pharmacological inhibition of Abl mitigates the neuronal defects caused by increased Dscam expression in vivo. (A) Nilotinib inhibits *Drosophila* Abl kinase. S2 cells were transfected with either Myc-vector or Abl::Myc, and then treated with either vehicle (DMSO) or 5 µM nilotinib for 6 hr. Total lysates were subjected to western blot analysis with phospho-Y412-Abl (p-Abl) (top) and Myc antibodies (bottom). (B) Quantification of the presynaptic terminal length of the indicated genotypes and drug treatment. Sample number is shown inside each bar. (C–H) Nilotinib treatment mitigates presynaptic arbor enlargement caused by Dscam overexpression (OE Dscam, D and E) and by *dFMRP* mutations (*dFMRP^Δ50M^*, G and H). Nilotinib treatment alone does not affect presynaptic terminal growth (F). The arrowhead in each panel points to the location where an axon elaborates the presynaptic terminal arbor. The MARCM technique was used to generate and visualize single presynaptic terminals of mutant C4da neurons. *Drosophila* larvae were raised in the presence of either 380 µM nilotinib or vehicle (DMSO) for 4 days before the analysis. Scale bar is 10 µm.

The following figure supplements are available for figure 4:

**Figure supplement 1**. Nilotinib and bafatinib do not reduce Dscam transgene expression.

**Figure supplement 2**. Nilotinib treatment does not cause defects in dendritic development or adult viability.

**Figure supplement 3**. Nilotinib and bafetinib act through Abl inhibition to mitigate Dscam-induced presynaptic arbor enlargement in vivo.

## Materials and methods

### Fly strains

abl[1] (*Gertler et al., 1989*), abl[4] (*Bennett and Hoffmann, 1992*), ppk-Gal4 (*Kuo et al., 2005*), UAS-Dscam[3.36.25.2]::GFP (*Yu et al., 2009*), UAS-Abl, UAS-BCR-Abl, UAS-Abl-K417N (*Wills et al., 1999b*), and dFMRP[Δ50M] (*Zhang et al., 2001*) were used in this study.

### DNA constructs and generation of transgenic flies

To generate pUASTattB-Abl::Myc for expression in S2 cells, the coding region of Abl was recovered from UAS-Abl transgenic flies by PCR, subcloned into pUASTattB-Myc by using the InFusion cloning system following manufacturer's protocol (Clontech, Mountain View, California). We generated pUASTattB-Abl-K417N::Myc by PCR mutagenesis as previously described (*O'Donnell and Bashaw, 2013*) from pUASTattB-Abl::Myc. UAS-Dscam[3.36.25.2]::GFP was previously generated as described (*Kim et al., 2013*). To generate UAS-DscamΔCyto, the Dscam coding region was digested with SstI and ligated with the GFP cDNA. Pickles2.31 was generously provided by Dr Yusuke Ohba at RIKEN Brain Science Institute (*Mizutani et al., 2010*). To generate UAS-Pickles2.31, the Pickles2.31 coding region was subcloned from pCAGGS-Pickles2.31 into pUASTattB using the InFusion cloning system following the manufacturer's protocol (Clontech). Transgenic flies carrying UAS-DscamΔCyto, UAS-Abl::Myc, and UAS-Pickles2.31 were generated by germline transformation with support from BestGene, Inc.

### Labeling presynaptic terminals using MARCM

The MARCM technique was used to visualize single neurons homozygous for abl[1], abl[4], or dFMRP[Δ50], and overexpressing Dscam[3.36.25.2]::GFP as previously described (*Kim et al., 2013*).

### Immunostaining and imaging

Immunostaining of third-instar larvae was accomplished as previously described (*Ye et al., 2011*). Antibodies used include chicken anti-GFP (Aves, Tigard, Oregon) and rabbit anti-RFP (Rockland, Limerick, Pennsylvania). Samples were dehydrated and mounted with DPX mounting media (Electron Microscopy Sciences, Hatfield, Pennsylvania). Confocal imaging was completed with a Leica SP5 confocal system equipped with a resonant scanner and 63× oil-immersion lens (NA = 1.40). Images were collected and quantified as previously described (*Kim et al., 2013*).

### S2 cell culture and transfection

*Drosophila* S2 cells were maintained in *Drosophila* Schneider's medium supplemented with 10% fetal bovine serum at 25°C in a humidified chamber. Cells were transfected with indicated DNA constructs together with tubulin-Gal4 (*Lee and Luo, 2001*) by using Lipofectamine 2000 (Life Technologies, Grand Island, New York) according to manufacturer's protocol.

### Co-immunoprecipitation and Western blotting

To perform co-immunoprecipitation, transfected S2 cells were harvested and lysed on ice with lysis buffer (50 mM Tris-HCl/pH 7.4, 150 mM NaCl, 2 mM sodium vanadate, 10 mM sodium fluoride, 1% Triton X-100, 10% glycerol, 10 mM imidazole and 0.5 mM phenylmethylsulfonyl fluoride). Lysates were centrifuged for 15 min at 20,000×*g*, 4°C and the resulting supernatant was incubated with Protein A/G PLUS-Agarose beads (Santa Cruz Biotechnology, Paso Robles, California) conjugated to mouse monoclonal anti-GFP clone 20 (Sigma-Aldrich, St. Louis, Missouri) for 4 hr at 4°C. After washing once with lysis buffer, twice with lysis buffer containing 0.1% deoxycholate, and 3 times with lysis buffer lacking Triton X-100, the immunoprecipitates and total lysates were resolved on 7.5% SDS-PAGE gels followed by western blot analysis as previously described (*Kim et al., 2013*).

Primary antibodies used in western blotting were mouse monoclonal anti-tubulin (Sigma), mouse anti-Myc (Sigma-Aldrich), mouse monoclonal anti-*Aequorea Victoria* GFP JL-8 (Clontech), and rabbit anti-phospho-Tyr412-c-Abl (Cell Signaling, Beverly, Massachusetts).

### In vivo Abl activity assay with Pickles2.31

To assay in vivo Abl activation, UAS-Pickles2.31 was expressed specifically in C4da neurons using ppk-Gal4 along with other UAS transgenes. The CNS was dissected from third-instar larvae into ice-cold

PBS with 2 mM sodium vanadate (~100 per experimental condition). After a brief centrifugation, larval CNSs were transferred into lysis buffer as described above in immunoprecipitation and western blotting. Cells were disrupted using a pestle followed by brief sonication. Immunoprecipitation and western blotting of Pickles2.31 was then accomplished as described above. Primary antibodies used were rabbit anti-eGFP (a gift from Dr Yang Hong) and rabbit anti-phospho-Tyr 207-CrkL (Cell Signaling).

## Drug treatment of *Drosophila* larvae and S2 cells

Nilotinib (Abcam, United Kingdom) and bafetinib (ApexBio Technology, Houston, Texas) were dissolved in dimethyl sulfoxide (DMSO) at 94 mM and 50 mM, respectively, as stock solutions before adding to S2 cells or fly food. S2 cells transfected with Abl::Myc were treated with either 5 µM nilotinib or the same volume of DMSO as a vehicle control for 6 hr before harvested and subjected to western blot analysis.

Nilotinib and bafetinib were administered to larvae by rearing the larvae on standard corn meal food containing different concentrations of the drugs. The highest concentrations that did not affect overall larval development were used. Fly viability on nilotinib treatment was performed by counting the number of adult flies. Seven virgin female and seven male flies were crossed and transferred to standard corn meal food containing either 380 µM nilotinib or the same volume of DMSO (0.4% final concentration). Embryos were collected for 24 hr and allowed to develop. Eclosed adult flies were counted on a daily basis.

The MARCM technique was used to generate and visualize mutant single C4da neurons as described above except that *Drosophila* embryos were collected and raised for 4 days on standard corn meal food containing either 380 µM nilotinib, 125 µM Bafetinib, or 0.4% DMSO. Sample preparation, imaging, and quantification were then completed as described above.

## Colocalization analysis

Colocalization of Dscam and Abl was quantified with Manders' Correlation Coefficients using the Just Another Colocalization Plugin (JACoP) (*Bolte and Cordelieres, 2006*) in ImageJ. Images were analyzed in three dimensions. Manders' Correlation Coefficients vary between 0 and 1, with 0 representing no overlap between images and 1 representing complete colocalization. $M_1$ and $M_2$ describe the overlap of each channel with the other (*Bolte and Cordelieres, 2006*). $M_1$ presents a measure of the fraction of Abl::Myc that overlaps Dscam(ΔCyto)::GFP, while $M_2$ presents a measure of the fraction of Dscam(ΔCyto)::GFP that overlaps Abl::myc.

## Statistical analysis

Two-way student's t- test was used for statistical analysis. *: $p < 0.05$; **: $p < 0.01$; ***: $p < 0.001$; ****: $p < 0.0001$; ns: not significant.

## Acknowledgements

We thank Dr Tzumin Lee for Dscam fly stocks, Dr Yusuke Ohba for the Pickles2.31 cDNA, Dr Yang Hong for the rabbit anti-GFP antibody, and Drs Cheng-yu Lee and Hideyuki Komori for the pUASTattB-Myc vector. This work was supported by grants from NIH (R01MH091186), Protein Folding Disease Initiative of the University of Michigan, and the Pew Scholars Program in the Biological Sciences to BY.

## Additional information

#### Competing interests

GRS: have a patent application on the use of Abl inhibitors as a therapeutic approach for treating brain disorders associated with dysregulated Dscam levels (Application number: PCT/US2014/072083). JHK: have a patent application on the use of Abl inhibitors as a therapeutic approach for treating brain disorders associated with dysregulated Dscam levels (Application number: PCT/US2014/072083). BY: have a patent application on the use of Abl inhibitors as a therapeutic approach for treating brain disorders associated with dysregulated Dscam levels (Application number: PCT/US2014/072083).

## Funding

| Funder | Grant reference | Author |
|---|---|---|
| National Institutes of Health (NIH) | R01MH091186 | Bing Ye |
| Pew Charitable Trusts | 2009-000359-016 | Bing Ye |
| University of Michigan (U-M) | Protein Folding Disease Initiative | Bing Ye |

The funders had no role in study design, data collection and interpretation, or the decision to submit the work for publication.

## Author contributions

GRS, Conceived the project and designed the experiments, Performed the Abl and Dscam overexpression experiments, the mosaic Abl loss-of-function experiments, the dendritic crossing experiments, the colocalization experiments, and the Abl binding and activation biochemistry experiments, Wrote the paper; JHK, Conceived the project and designed the experiments, Performed the pilot experiments on Abl overexpression, and performed the pharmacological studies, Wrote the paper; BY, Conceived the project and designed the experiments, Supervised the project, Wrote the paper

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
