## [Decision Letter]

Thank you for sending your work entitled “Dysregulated Dscam levels act through Abelson tyrosine kinase to enlarge presynaptic arbors” for consideration at *eLife*. Your article has been favorably evaluated by K VijayRaghavan (Senior editor), a Reviewing editor, and three reviewers.

The Reviewing editor and the reviewers discussed their comments before we reached this decision, and the Reviewing editor has assembled the following comments to help you prepare a revised submission.

We note key findings of the paper which show that loss of Abl kinase can reduce presynaptic terminal overgrowth caused by DsCAM overexpression, DsCAM intracellular domain can interact and activate Abl kinase and in particular that an FDA approved Abl inhibitor nilotinib can reduce presynaptic overgrowth caused by DsCAM overexpression or in fly *dFMRP* mutants, which also show increased Dscam expression. An added significance of the work therefore is the potential translational application of Abl inhibitors in treatment of neurological disorders.

General assessment:

CNS phenotypes for *Dscam* loss or misexpression had been explored in prior work. [2] showed that loss of both Dscam and its paralog Dscam3 result in CNS midline crossing defects, and that overexpression of Dscam results in ectopic midline crossings. [43] showed that RNAi knockdown of Dscam resulted in morphogenesis defects as well as commissureless phenotypes. However, the effects of *Dscam* loss at axon terminals growth does not appear to have been demonstrated. Overall, this is a well-written and interesting study with some exciting results linking Dscam to a key axonal signalling molecule, even if some aspects of the work are at a preliminary stage.

The conclusion that Abl kinase activity is a relevant output of Dscam in determining presynaptic arbor size is supported by a convincing set of experiments in Figure 1 (and supplements), and this is a real strength of the manuscript. Given how little is known about signal transduction downstream of Dscam, this is significant. Further, the observation that Dscam signaling in regulating presynaptic arbor size and dendrite self-avoidance involves different molecular mechanisms is an interesting discovery. The studies demonstrating that Dscam can interact with Abl and promote Abl activation in S2 cells support the model that Dscam modulates Abl activity in neurons, but demonstrating the interaction in neurons would substantially improve the manuscript. Finally, the authors convincingly show that the Abl pharmacological inhibitor nilotinib can suppress presynaptic arbor overgrowth induced by Dscam or FMRP mutation, but the mechanism of action is unclear. The following substantive concerns need to be addressed.

Substantive concerns:

1a) A principal concern is that data for in vivo specificity of nilotinib on Abl are not convincing. Figure 3 is not interpretable without controls, such as *abl(lf)* and *abl(CA)* or DsCAM overexpression. Figure 3, the analyses on the neuronal phenotypes in flies fed with nilotinib are surprisingly limited or selective. The value of using a fly model to test potential drug targets is that we can gain a full understanding of any possible off-target or side effects. Thus, it is equally important that they show other aspects of the neurons and animal health etc. are, or are not, affected by the drug treatment. Moreover, one can't find any data showing how the dose of nilotinib treatment, which would be highly relevant for any further interests in translational application, was decided.

1b) The specificity of the inhibitor is not discussed (or examined). Nilotinib inhibits a number of tyrosine kinases and appears to broadly inhibit tyrosine phosphorylation in larvae. The assertion that the effects on presynaptic arbor size reflect inhibition of Abl activation is not convincing. Do other small molecule inhibitors of Abl exist that have a more targeted effect (i.e., they don't broadly block tyrosine kinases) and if so, do they similarly affect presynaptic arbor growth?

1c) Figure 3. The specificity of nilotinib to Abl in *Drosophila* is not verified. The decrease in phospho-tyrosine in 3b demonstrates a general decrease in the CNS, and does not necessarily show a specific targeting of Abl. This must be addressed. To demonstrate specificity, a possible experiment is to repeat the experiment shown in 3B in flies that are mutated in the Abl phosphorylation site Tyr412, or simply null for Abl. If nilotinib is specific to blocking phospho-Abl, there should be no difference in p-Tyr levels whether or not nilotinib is added, if Abl phospho-sites are blocked or Abl is missing. However, if addition of nilotinib still results in a decrease in p-Tyr even when Abl phospho-sites are blocked, this suggests nilotinib is having an effect on the catalytic activity of other kinases.

2a) Figure 3. What is the effect of nilotinib treatment in wild-type flies? The results of wild type + nilotinib can be shown in addition to the experiments in 3c-g. It would be very reassuring to see that nilotinib elicits *abl*-like phenotypes in other parts of the nervous system or elsewhere (e.g. epithelial development in the embryo)—if the drug is selectively blocking Abl, it should phenocopy *abl* mutants, but not mutants in other kinases.

2b) What is the effect of drug treatment on presynaptic arbor size in a wild type background? It's impossible to evaluate the experimental results without this control.

3a) The data presented suggest that Dscam can associate with Abl and promote Abl phosphorylation, but this is not demonstrated in vivo.

3b) Figure 2. The results shown in panel 2B and 2C demonstrate binding of Abl to the cytoplasmic domain of Dscam in S2 cells using transfected constructs and levels of expression that may contribute to detection of non-physiological interaction. This is important because the fraction of the input pool that IPs with Dscam in cells overexpressing both proteins is rather low. More importantly, they do not show that this binding occurs in vivo, and that even if it does occur in vivo, that this binding is important to axon terminal growth in particular. To determine the role of Abl-Dscam binding in vivo the group should consider performing fluorescence tagging and/or immunolocalization of Abl in the three genetic backgrounds shown in panel 2A (WT, Dscam::GFP OE, Dscam cyto::GFP OE). If there is a bona fide interaction with Dscam at the terminal, co-localization of Abl with Dscam::GFP should be observed. Moreover, if this interaction requires Dscam cytoplasmic domain, expression of Dscam cyto::GFP instead of full- length Dscam should result in a re-distribution of Abl.

4) Figure 1. The text (second paragraph of the Results section) and the figure legend state that Abl overexpression leads to a pre-synaptic overgrowth phenotype. However, while a defect in the axons is apparent in panel 1B, it is not clear whether this is due to increased growth necessarily. For example, the gross defects could result from fasciculation defects between axons as they enter the CNS neuropil. Indeed, the apparent decrease in mRFP signal intensity in the Abl OE flies raise the possibility that it is not overgrowth, but defasiculation, that is observed in 1B. This is important because Abl has been linked to many receptors/adhesion molecules that may also contribute to the normal neuropil structure for sensory terminals. Abl nulls are not shown to confirm that the phenotypes are only selective to overexpression.

5) Figure 1, general comment. All axon terminal phenotypes shown are for Dscam OE. While *Dscam* loss phenotypes have been shown in other systems (e.g. midline crossing), it has not been demonstrated for terminal growth. It would be useful to include data on the effects of *Dscam* loss at the synaptic terminal, as well as genetic experiments on the interaction between *Dscam* and *abl* loss of function alleles (as has been done in other systems). If the two genes are in one pathway for sensory terminal growth, then the double null should be indistinguishable from the single mutants. It is curious that Abl mutant terminals look relatively normal. Does this mean that Abl is not required for other receptors or DSCAM under normal expression levels?

6) Figure 1—figure supplement 1. Dscam-GFP levels in the neuronal soma is shown to be unchanged in *abl* mutants. However, the observations in the cell body do not rule out the possibility that at the synapse, *abl* loss could have effects on Dscam levels, perhaps due to changes in Dscam protein trafficking (Abl has been implicated in axon transport by Bill Saxton). Just because *abl* nulls revert the Dscam OE phenotype, this doesn't prove a downstream function. Abl can be required upstream of Dscam at the synapse. To test this, a parallel analysis of Dscam-GFP could be done at axon terminals to verify that the effect of *abl* is the same both in the soma and at the presynaptic terminal.

7) The *abl C4da* dendritic phenotype analysis should be added to the supplemental figures.

8) Figure 2. There are no data confirming the stability of the Dscam cytoplasmic domain deletion. In fact the recovery in the IP (Figure 2) is very poor and raises the concern that this receptor mutant is not capable of accumulating to the same concentration in the OE experiment, which could explain the lack of an in vivo OE phenotype in a way that does not support requirement of the Cyto domain. Moreover, why is full-length Dscam detected as a doublet? Can this be explained by cleavage, phosphorylation, etc.?

---

## [Author Response]

*1a) A principal concern is that data for in vivo specificity of nilotinib on Abl are not convincing.*
Figure 3
*is not interpretable without controls, such as* abl(lf) *and* abl(CA) *or DsCAM overexpression.*
Figure 3*, the analyses on the neuronal phenotypes in flies fed with nilotinib are surprisingly limited or selective. The value of using a fly model to test potential drug targets is that we can gain a full understanding of any possible off-target or side effects. Thus, it is equally important that they show other aspects of the neurons and animal health etc. are, or are not, affected by the drug treatment. Moreover, one can't find any data showing how the dose of nilotinib treatment, which would be highly relevant for any further interests in translational application, was decided*.

We appreciate the reviewers’ critiques. Following up on the reviewers’ suggestions, we have investigated the in vivo specificity of nilotinib on Abl with several experiments. The results strongly support that the effect of nilotinib on Dscam-induced presynaptic arbor growth is through Abl. In addition, we have addressed the issue of off-targets in our response to comment #1b). In the initial submission, we presented Figure 3 to show the accessibility of nilotinib into central nervous system in fly, rather than to show the specificity of nilotinib. We apologize for the confusion, and have removed the original Figure 3 to avoid the confusion.

In the revised manuscript, we have included the analysis of the viability of flies fed either vehicle (DMSO) or nilotinib (new Figure 4—figure supplement 2), which shows no difference between the two groups. Furthermore, we have analyzed C4da dendrite development in flies treated with nilotinib and found no difference between DMSO- and nilotinib-treated larvae (new Figure 4—figure supplement 2). These results suggest that the nilotinib concentration required for rescuing axonal phenotypes does not cause any severe side effects. Together with known clinical safety of nilotinib (Kantarjian et al 2011. nilotinib is effective in patients with chronic myeloid leukemia in chronic phase after imatinib resistance orintolerance: 24-month follow-up results Blood 117 (4), 1141-1145), these results show the therapeutic potential of nilotinib.

In the revised manuscript (in the subsection headed “Drug treatment of *Drosophila* larvae and S2 cells”), we have explained how the dose of nilotinib (and bafetinib) for treating larvae was selected. The dose was the highest concentration (among various concentrations tested) that was tolerated by the larvae without displaying any toxic responses.

1b) The specificity of the inhibitor is not discussed (or examined). Nilotinib inhibits a number of tyrosine kinases and appears to broadly inhibit tyrosine phosphorylation in larvae. The assertion that the effects on presynaptic arbor size reflect inhibition of Abl activation is not convincing. Do other small molecule inhibitors of Abl exist that have a more targeted effect (i.e., they don't broadly block tyrosine kinases) and if so, do they similarly affect presynaptic arbor growth?

The reviewers are correct that, depending on the concentration, nilotinib also inhibits several other targets. We have added a discussion on nilotinib specificity in the revised manuscript (pleas see the Results and Discussion section). In addition, we tested another Abl inhibitor, bafetinib, which has a chemical scaffold different from nilotinib. This compound is also known to cross blood-brain barrier in mice. We found that the administration of bafetinib, like nilotinib, was able to significantly rescue axon terminal length in larvae overexpressing Dscam (Figure 4—figure supplement 3). The known off-targets of nilotinib in mammals are c-Kit, PDGFR, Arg, NQ02, and DDR1, while those of bafetinib are Fyn and Lyn. Since the off-targets of these two drugs are different, their common effect on Dscam-induced axon terminal growth is likely through Abl inhibition. Furthermore, we treated the larvae overexpressing Dscam in homozygous *abl*^-/-^ neurons and found that the nilotinib treatment did not further decrease size of presynaptic arbors (Figure 4—figure supplement 3). Taken together, even though nilotinib also inhibits several other tyrosine kinases, its effect of rescuing enlarged presynaptic arbors is through Abl inhibition.

*1c)*
Figure 3*. The specificity of nilotinib to Abl in Drosophila is not verified. The decrease in phospho-tyrosine in 3b demonstrates a general decrease in the CNS, and does not necessarily show a specific targeting of Abl. This must be addressed. To demonstrate specificity, a possible experiment is to repeat the experiment shown in 3B in flies that are mutated in the Abl phosphorylation site Tyr412, or simply null for Abl. If nilotinib is specific to blocking phospho-Abl, there should be no difference in p-Tyr levels whether or not nilotinib is added, if Abl phospho-sites are blocked or Abl is missing. However, if addition of nilotinib still results in a decrease in p-Tyr even when Abl phospho-sites are blocked, this suggests nilotinib is having an effect on the catalytic activity of other kinases*.

We again apologize for the confusion caused by previous Figure 3, which was to evaluate the accessibility of nilotinib into central nervous system in fly rather than the specificity of nilotinib treatment. In agreement with that nilotinib has off-targets, we found that Abl inhibition is not the main cause of the decrease in overall tyrosine phosphorylation in larval brain upon nilotinib treatment. The extent of the decrease in overall tyrosine phosphorylation persists even when nilotinib was fed to homozygous *Abl*^-/-^ mutants (data not shown).

Nevertheless, as we explained in the response to comment #1b), we have performed additional experiments to demonstrate the rescuing effect of nilotinib on Dscam-induced overgrowth of presynaptic arbors is due to the inhibition of Abl. We have also added a discussion of nilotinib specificity in the revised manuscript (please see the Results and Discussion section).

*2a)*
Figure 3*. What is the effect of nilotinib treatment in wild-type flies? The results of wild type + nilotinib can be shown in addition to the experiments in 3c-g. It would be very reassuring to see that nilotinib elicits* abl*-like phenotypes in other parts of the nervous system or elsewhere (e.g. epithelial development in the embryo)—if the drug is selectively blocking Abl, it should phenocopy* abl *mutants, but not mutants in other kinases*.

Although Abl is involved in multiple processes in embryonic development, we do not expect to observe similar severity of phenotypes in nilotinib-treated animals as *abl* mutant. This is because nilotinib was administrated through food and was thus not accessible to the animal until embryonic development is completed. Moreover, the nilotinib treatment in our experiments likely partially inhibited Abl activity for three reasons. First, the efficacy of presynaptic arbor rescue in Dscam overexpressing neurons was partial as compared to the complete rescue achieved by homozygous *Abl*^*-/-*^ mutations (compare new Figure 4B-G with new Figure 1). Second, the same dose of nilotinib almost completely rescued presynaptic arbor enlargement in *FMRP*^*-/-*^ mutants while reducing only 50% of *Abl* gene dosage was sufficient for the same extent of rescue (compare new Figure 4 with new Figure 1). Third, homozygous mutation of *abl* causes pupal lethality (or embryonic lethality when *abl* is mutated maternally) while nilotinib treatment at the concentration used does not affect fly development through adulthood (Figure 4—figure supplement 2). Finally, Abl is known to have kinase-independent functions ([16]. A novel tyrosine kinase-independent function of *Drosophila abl* correlates with proper subcellular localization. Cell 63:949-960), so even if nilotinib were capable of inhibiting all Abl kinase activity, we would not expect to see identical phenotypes as *abl* mutants.

We have performed additional experiments whose results support the conclusion that the effects of nilotinib on presynaptic arbor growth is through Abl inhibition (see the response to comment #1b). Our new results also demonstrate that the nilotinib concentration used in this study does not affect neuronal and animal development (see the response to comment #1a). Therefore, although nilotinib does not only inhibit Abl, its inhibition of Abl kinase is responsible for mitigating the presynaptic terminal overgrowth caused by dysregulated Dscam.

*2b) What is the effect of drug treatment on presynaptic arbor size in a wild type background? It's impossible to evaluate the experimental results without this control*.

We have performed the experiment and added in new Figure 4. There is no difference in axon terminal length between wild-type flies treated with vehicle (DMSO) and with nilotinib (for the statistics see new Figure 4).

*3a) The data presented suggest that Dscam can associate with Abl and promote Abl phosphorylation, but this is not demonstrated in vivo*.

We have performed two additional experiments to address this concern. The new results are presented in the new Figures 2 and 3.

First, to test if Dscam and Abl associate in vivo we expressed Dscam::GFP or DscamΔCyto::GFP along with Abl::Myc in C4da neurons, and then analyzed the extent that Dscam and Abl::Myc colocalized in C4da axon terminals (new Figure 2). These results support the notion that Abl binds to Dscam in vivo in axon terminals.

Second, to determine whether Dscam promotes Abl phosphorylation in vivo, we developed a novel method that reports Abl activation only in the axon terminals of C4da neurons in vivo (new Figure 3). We specifically expressed a probe that reports Abl activity in C4da neurons. Since the cell bodies of C4da neurons reside in the body wall, in the larval CNS this probe is only present in the axon terminals of C4da neurons. The Abl activity probe was then immunoprecipitated from the larval CNS after dissection. The phosphorylation state of the Abl target site in the probe was monitored with western blotting by using a phospho-specific antibody. Compared to a negative control transgene, expression of Dscam led to an increase in probe phosphorylation while DscamΔCyto did not (new Figure 3), suggesting that Dscam activates Abl in vivo and that this activation requires the Dscam cytoplasmic domain.

*3b)*
Figure 2*. The results shown in panel 2B and 2C demonstrate binding of Abl to the cytoplasmic domain of Dscam in S2 cells using transfected constructs and levels of expression that may contribute to detection of non-physiological interaction. This is important because the fraction of the input pool that IPs with Dscam in cells overexpressing both proteins is rather low. More importantly, they do not show that this binding occurs in vivo, and that even if it does occur in vivo, that this binding is important to axon terminal growth in particular. To determine the role of Abl-Dscam binding in vivo the group should consider performing fluorescence tagging and/or immunolocalization of Abl in the three genetic backgrounds shown in panel 2A (WT, Dscam::GFP OE, Dscam cyto::GFP OE). If there is a bona fide interaction with Dscam at the terminal, co-localization of Abl with Dscam::GFP should be observed. Moreover, if this interaction requires Dscam cytoplasmic domain, expression of Dscam cyto::GFP instead of full- length Dscam should result in a re-distribution of Abl*.

Please refer to our response to comment #3a) above.

*4)*
Figure 1*. The text (second paragraph of the Results section) and the figure legend state that Abl overexpression leads to a pre-synaptic overgrowth phenotype. However, while a defect in the axons is apparent in panel 1B, it is not clear whether this is due to increased growth necessarily. For example, the gross defects could result from fasciculation defects between axons as they enter the CNS neuropil. Indeed, the apparent decrease in mRFP signal intensity in the Abl OE flies raise the possibility that it is not overgrowth, but defasiculation, that is observed in 1B. This is important because Abl has been linked to many receptors/adhesion molecules that may also contribute to the normal neuropil structure for sensory terminals. Abl nulls are not shown to confirm that the phenotypes are only selective to overexpression.*

In the abdominal segments 5-8 (A5-A8), the axon terminals of the three C4da neurons in each hemi-segment consist of an anterior projection that extends within one segment length. These axon terminals form a fascicle that connects two adjacent neuropils. Between the axon entry points of abdominal segment 5 and 6, fewer than three connectives were typically observed (new Figure 1) (Wang, et al., 2014, PLOS Biology 11 (6). e1001572). Both BCR-Abl and Abl overexpression caused thickened bundle of connectives in the C4da neuropil ladder (new Figure 1). We have quantified the number of connectives between the neuropil of abdominal segment 5(A5) and that of A6 (new Figure 1). The results clearly showed an overgrowth phenotype as opposed to a defasciculation phenotype because overexpression of Abl/BCR-Abl led to a significant increase in the number of connectives when compared to wild-type. The increase in the number of connectives could either arise from increased axon branches from neurons in the same segment or overextended axons from other segments.

This particular assay is only useful for analyzing striking changes in axon growth, as single axon terminals cannot be distinguished. *Abl* null mutants were not shown since the decrease in axon terminal growth is subtle and would not be visible when all C4da axon terminals are imaged together. We have demonstrated the effects of *Abl* null mutations using single-cell genetic mosaic analysis (new Figure 1), and the statistical analysis of the results is shown in the new Figure 1.

*5)*
Figure 1*, general comment. All axon terminal phenotypes shown are for Dscam OE. While* Dscam *loss phenotypes have been shown in other systems (e.g. midline crossing), it has not been demonstrated for terminal growth. It would be useful to include data on the effects of* Dscam *loss at the synaptic terminal, as well as genetic experiments on the interaction between* Dscam *and* abl *loss of function alleles (as has been done in other systems). If the two genes are in one pathway for sensory terminal growth, then the double null should be indistinguishable from the single mutants. It is curious that Abl mutant terminals look relatively normal. Does this mean that Abl is not required for other receptors or DSCAM under normal expression levels?*

We have demonstrated the consequences of *Dscam* loss of function in C4da axon terminal growth in a previous paper ([19]. Dscam expression levels determine presynaptic arbor sizes in *Drosophila* sensory neurons. Neuron 78:827-38). Null alleles of *Dscam* almost completely block presynaptic arbor growth. However, the reviewers’ comments made us aware that a discussion of the results of our previous work would be helpful to understanding the current manuscript and we have added this in the Results and Discussion section. Analysis of Dscam and Abl double loss of function larvae is technically challenging since *Dscam* mutations are embryonic lethal. Moreover, MARCM analysis of two genes located on different chromosomes is extremely difficult and in fact has not been demonstrated before.

Abl is known to be maternally contributed to the embryonic development, and thus there is likely to be some Abl even in *abl* null embryos. We think that it is likely that Abl is required for normal presynaptic arbor growth. In fact, we observed a subtle but significant reduction in presynaptic arbor size *abl* MARCM clones (Figure 1). Nevertheless, our results demonstrate that the maternally contributed Abl is insufficient for the Dscam-induced presynaptic overgrowth caused by Dscam overexpression or genetic mutations of *FMRP*.

*6)*
Figure 1—figure supplement 1*. Dscam-GFP levels in the neuronal soma is shown to be unchanged in* abl *mutants. However, the observations in the cell body do not rule out the possibility that at the synapse,* abl *loss could have effects on Dscam levels, perhaps due to changes in Dscam protein trafficking (Abl has been implicated in axon transport by Bill Saxton). Just because* abl *nulls revert the Dscam OE phenotype, this doesn't prove a downstream function. Abl can be required upstream of Dscam at the synapse. To test this, a parallel analysis of Dscam-GFP could be done at axon terminals to verify that the effect of* abl *is the same both in the soma and at the presynaptic terminal*.

We agree that genetic epistasis does not prove an upstream-downstream relationship. In addition to the in vitro biochemistry studies suggesting that Dscam activates Abl, in the revised manuscript we show new results demonstrating that Dscam activates Abl in C4da axon terminals (new Figure 3). Prompted by the reviewers’ suggestion, we have analyzed the effects of *abl* loss of function on Dscam expression in axon terminals. Using single-cell genetic mosaic analysis, we found that Dscam expression was unchanged in *abl* loss of function clones (new Figure 1—figure supplement 1).

*7) The* abl C4da *dendritic phenotype analysis should be added to the supplemental figures*.

As requested, we have added supplemental figures to Figure 1 (Figure 1—figure supplement 2) showing that there is no difference in dendritic length between wild-type and *abl* mutant neurons using MARCM. We have also added supplemental figures to Figure 4 (Figure 4—figure supplement 2) showing that nilotinib does not affect dendritic development.

*8)*
Figure 2*. There are no data confirming the stability of the Dscam cytoplasmic domain deletion. In fact the recovery in the IP (*Figure 2*) is very poor and raises the concern that this receptor mutant is not capable of accumulating to the same concentration in the OE experiment, which could explain the lack of an in vivo OE phenotype in a way that does not support requirement of the Cyto domain. Moreover, why is full-length Dscam detected as a doublet? Can this be explained by cleavage, phosphorylation, etc*.*?*

Thank you for making us aware that this point needed clarification. We have analyzed the expression and presence of this *Dscam* mutant, which is GFP-tagged, in axon terminals. The new result is shown in the Figure 2—figure supplement 1. When stained and imaged simultaneously, Dscam-GFP and DscamΔCyto-GFP are present in the axon terminals at comparable levels. In addition, we saw robust axon terminal overgrowth in the Dscam condition but never saw any change in axon terminal growth in the DscamΔCyto condition.

We believe that the lower molecular weight band of the doublet in old Figure 2 is a degradation product. This is supported by the lack of this doublet under essentially identical transfection conditions, as shown in Figure 2.